# Strain-shear coupling in bilayer MoS$_2$

Jae-Ung Lee [1], Sungjong Woo[2], Jaesung Park[1], Hee Chul Park[2,3], Young-Woo Son[2] & Hyeonsik Cheong [1]

Layered materials such as graphite and transition metal dichalcogenides have extremely anisotropic mechanical properties owing to orders of magnitude difference between in-plane and out-of-plane interatomic interaction strengths. Although effects of mechanical perturbations on either intralayer or interlayer interactions have been extensively investigated, mutual correlations between them have rarely been addressed. Here, we show that layered materials have an inevitable coupling between in-plane uniaxial strain and interlayer shear. Because of this, the uniaxial in-plane strain induces an anomalous splitting of the degenerate interlayer shear phonon modes such that the split shear mode along the tensile strain is not softened but hardened contrary to the case of intralayer phonon modes. We confirm the effect by measuring Raman shifts of shear modes of bilayer MoS$_2$ under strain. Moreover, by analyzing the splitting, we obtain an unexplored off-diagonal elastic constant, demonstrating that Raman spectroscopy can determine almost all mechanical constants of layered materials.

[1] Department of Physics, Sogang University, Seoul 04107, Korea. [2] Korea Institute for Advanced Study, Seoul 02455, Korea. [3] Center for Theoretical Physics of Complex Systems, Institute of Basic Science, Daejon 34051, Korea. Jae-Ung Lee and Sungjong Woo contributed equally to this work. Correspondence and requests for materials should be addressed to Y.-W.S. (email: hand@kias.re.kr) or to H.C. (email: hcheong@sogang.ac.kr)

Three-dimensional (3D) layered materials such as graphite and transition metal dichalcogenides (TMDs) are formed by weak van der Waals force between two-dimensional (2D) crystals whereas atoms within each 2D crystal are bound through a strong covalent bonding. A huge difference between in-plane and out-of-plane interatomic interaction strengths often result in extremely anisotropic mechanical properties of 3D-layered systems. Since electronic and mechanical properties of multilayered 2D crystals critically depend on their interlayer interactions[1–3], understanding the influence of the external mechanical perturbations on the interlayer as well as intralayer interaction is very important. Mechanical properties of a single-layer 2D crystal can be understood by analyzing variations of high-frequency optical phonon modes with external mechanical perturbations[4–12]. Similarly, the effects of interlayer interactions on mechanical properties can be analyzed by probing the low-frequency interlayer shear and breathing modes[13–17]. Raman spectroscopy, an invaluable diagnostic tool for 2D crystals, have played an important role in measuring these modes for various layered materials such as graphene and TMDs[13–28]. For example, intralayer[11,12] and interlayer[13–17] elastic moduli as well as their variations due to rotational and translational stacking faults[29,30] can be obtained by examining the ultralow-frequency Raman spectrum of layered 2D materials. As introduced here, the effects of mechanical perturbations on physical properties related with either intralayer or interlayer interactions separately have been studied extensively whereas mutual interplay between them have rarely been studied.

In this study, we attempt to elucidate the overall effect of external mechanical strain on the interlayer and intralayer interactions, as well as the coupling between them. We first analyze the effect of tensile strain on bilayer $MoS_2$ using the analytic linear elastic model. Then, we perform first-principles calculations on elastic constants of bilayer $MoS_2$, a prototypical TMD material. Both the analytic model and the first-principles calculations predict that a uniaxial external mechanical strain along in-plane direction indeed couples to interlayer shear, splitting the degenerate low-energy shear phonon modes. This indicates relative sliding of two layers with an external strain along in-plane direction. Contrary to usual phonon softening under tensile strain, our calculations show that the softened shear mode in strained bilayer $MoS_2$ is perpendicular to the applied tensile strain direction and vice versa. Next, we confirm the predicted frequency shift and splitting of the shear modes of bilayer $MoS_2$ by carrying out ultralow-frequency Raman scattering measurements under uniaxial strain. From a careful analysis on the polarized Raman spectroscopy data on split shear phonon modes, we can measure the off-diagonal elastic constant of the bilayer system that has not yet been explored. Our results could therefore provide a way to measure a whole set of elastic constants of layered systems that characterize their mechanical properties completely.

## Results

**Coupling between uniaxial strain and shear.** The bilayer $2H$-$MoS_2$ has a stacked trigonal prismatic structure with inversion and three-fold rotational symmetry (Fig. 1a, b). So, the compliance tensor of the system with $D_{3d}$ symmetry in a matrix form[31] is given by

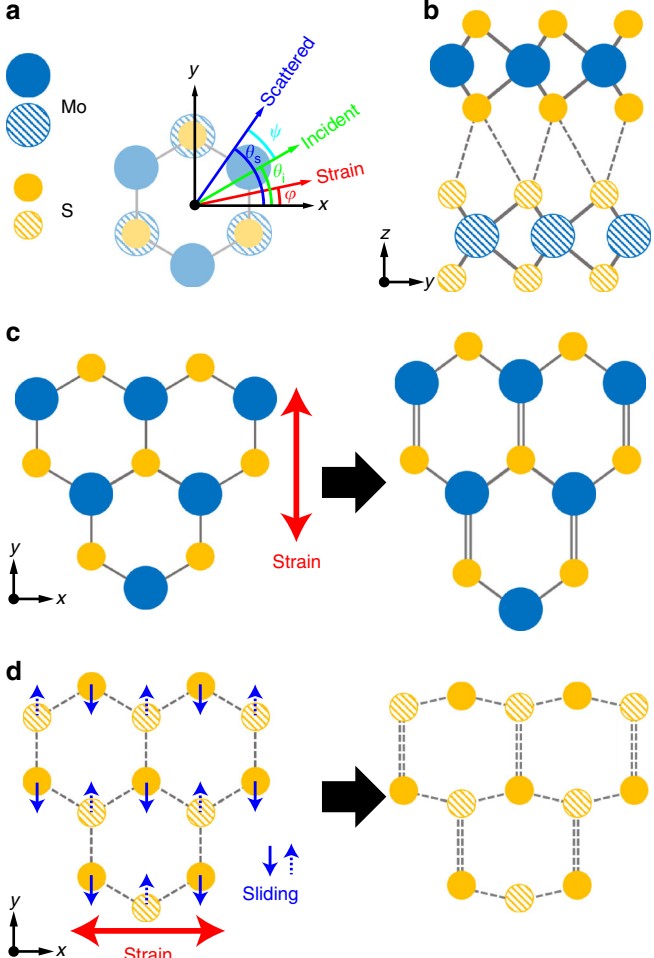

**Fig. 1** Lattice structure of bilayer $MoS_2$ under uniaxial strain. **a** Definitions of angles. **b** Side view of crystal structure of bilayer $MoS_2$. The atoms in the top layer are shown as solid circles, whereas those in the bottom layer are shown as hashed circles. **c** Top view of unstrained single-layer $MoS_2$ (left panel) and distorted lattice structure under armchair strain (right panel). Solid lines are for Mo–S covalent bonds. The double lines denote a longer bond distance compared to the single line. **d** Top view of hexagonal network of interface sulfur atoms in unstrained bilayer $MoS_2$ (left) and distorted lattice structure under zigzag strain (right). Sulfur atoms belonging to the upper (lower) layer are denoted by yellow filled (hashed) circles. The dotted lines indicate interlayer van der Waals bonds between interface sulfur atoms. Here, the double and single lines indicate the lengthened and shortened interfacial sulfur bond distances, respectively. Zigzag strain in single-layer $MoS_2$ and armchair strain in bilayer $MoS_2$ induce similar lattice distortions

$$
\begin{pmatrix} \varepsilon_1 \\ \varepsilon_2 \\ \varepsilon_3 \\ \varepsilon_4 \\ \varepsilon_5 \\ \varepsilon_6 \end{pmatrix} = \begin{pmatrix} 1/E_i & -v_i/E_i & -v_0/E_i & s_{14} & 0 & 0 \\ -v_i/E_i & 1/E_i & -v_0/E_i & -s_{14} & 0 & 0 \\ -v_0/E_i & -v_0/E_i & 1/E_0 & 0 & 0 & 0 \\ s_{14} & -s_{14} & 0 & s_{44} & 0 & 0 \\ 0 & 0 & 0 & 0 & s_{44} & -2s_{14} \\ 0 & 0 & 0 & 0 & -2s_{14} & 2(1+v_i)/E_i \end{pmatrix} \begin{pmatrix} \sigma_1 \\ \sigma_2 \\ \sigma_3 \\ \sigma_4 \\ \sigma_5 \\ \sigma_6 \end{pmatrix}.
$$

$$(1)$$

Here, $E_{i(o)}$ is the in-plane (out-of-plane) Young's modulus, $v_{i(o)}$ is the in-plane (out-of-plane) Poisson's ratio, and the vector form of strain and stress are defined as $\varepsilon_1 = \varepsilon_{xx}$, $\varepsilon_2 = \varepsilon_{yy}$, $\varepsilon_3 = \varepsilon_{zz}$, $\varepsilon_4 = 2\varepsilon_{yz}$, $\varepsilon_5 = 2\varepsilon_{xz}$, $\varepsilon_6 = 2\varepsilon_{xy}$, $\sigma_1 = \sigma_{xx}$, $\sigma_2 = \sigma_{yy}$, $\sigma_3 = \sigma_{zz}$, $\sigma_4 = \sigma_{yz}$, $\sigma_5 = \sigma_{xz}$, and $\sigma_6 = \sigma_{xy}$. The tensor form of strain is further defined as $\varepsilon_{ij} = 1/2(\partial u_i/\partial x_j + \partial u_j/\partial x_i)$ with $u_i$ being the local displacement vector component, and the corresponding stress tensor is $\sigma_{ij} = \partial F/\partial \varepsilon_{ij}$, where $F$ is the free energy of the system[31]. A constant uniaxial tensile stress $\sigma$ along a direction at an angle $\varphi$ from the $x$-axis in Fig. 1a corresponds to a stress tensor with components $\sigma_1 = \sigma \cos^2\varphi$, $\sigma_2 = \sigma \sin^2\varphi$, $\sigma_6 = \sigma \cos\varphi \sin\varphi$, and $\sigma_3 = \sigma_4 = \sigma_5 = 0$,

using rotational tensor transformation of $\sigma_{ij}$. From Eq. (1), we find that such a stress leads to strain components $\varepsilon_4 = s_{14}\sigma \sin(\pi/2 - 2\varphi)$ and $\varepsilon_5 = s_{14}\sigma \cos(\pi/2 - 2\varphi)$. These correspond to lateral sliding of the upper layer with respect to the lower layer by $\Delta r = (\Delta x, \Delta y)$ with $\Delta x = 2d_{\text{int}}\varepsilon_{xz} = d_{\text{int}}\varepsilon_5 = d_{\text{int}}s_{14}\sigma \cos(\pi/2 - 2\varphi)$ and $\Delta y = 2d_{\text{int}}\varepsilon_{yz} = d_{\text{int}}\varepsilon_4 = d_{\text{int}}s_{14}\sigma \sin(\pi/2 - 2\varphi)$. This implies that the uniaxial stress causes lateral sliding of the upper layer with respect to the lower layer in the direction at an angle of $-2\varphi$ from the positive $y$-axis if $s_{14} > 0$. If $s_{14} < 0$, the direction of sliding is reversed toward the angle $-2\varphi$ from the negative $y$-axis, which is $\pi - 2\varphi$ from the positive $y$-axis.

For a simple case of a constant uniaxial tensile stress $\sigma$ along the $x$-direction ($\varphi = 0°$) as shown in Fig. 1d, $\sigma_1 = \sigma$ and $\sigma_2 = \ldots = \sigma_6 = 0$, which lead to an off-diagonal strain component $\varepsilon_4 = s_{14}\sigma$. This indicates that the stress can induce a lateral sliding between the two layers given by $\Delta y = d_{\text{int}}\varepsilon_4 = d_{\text{int}}s_{14}\sigma$, where $d_{\text{int}}$ is the interlayer distance.

**First-principles calculations of shear mode splitting.** Applying a tensile strain typically softens the phonons along the direction of the strain because the effective spring constant along it becomes weaker owing to the elongation of interatomic distances[4,6–10,32]. Such a trend is found in our first-principles calculations and other studies for the intralayer optical phonon modes of strained $MoS_2$[7–10]. For the intralayer $E_g$ mode of bilayer $MoS_2$, the calculated frequency shifts match our experiments very well and are found to be linear up to 2% strain (see Supplementary Fig. 1). However, opposite results are obtained for the interlayer shear modes. Figure 2 shows the calculated phonon frequency shift of the interlayer shear modes of the same system with respect to the uniaxial strain; the higher frequency mode ($S^+$) corresponds to the vibrations along the strain axis, whereas the lower frequency mode ($S^-$) corresponds to the vibrations perpendicular to it. Our first-principles calculations show that the shifts barely depend on the strain direction (see Fig. 2 and Supplementary Fig. 2). By inspecting the relaxed atomic structures for phonon calculation under strain, we have found that the two layers slide with respect to each other as uniaxial strain is applied. This indeed agrees with the aforementioned analysis of the compliance tensor of the system. The amount of strain-induced interlayer sliding based on the first-principles calculation is not negligible at all and is about a half percent of the lattice constant for the applied strain of 1%. It is also confirmed that the direction of sliding matches well with our model prediction.

We have found that the stress-induced layer sliding is the origin of the anomalous shear mode splitting. We first assume that the interlayer coupling for the shear modes is mainly determined by the interaction between adjacent sulfur atoms at the interface. These interface sulfur atoms also form a buckled hexagonal lattice, where each sublattice belongs to a different sulfur layer (Fig. 1d). Therefore, the splitting of shear modes would solely depend on the way of breaking the hexagonal symmetry in the hexagonal network between the nearest sulfur atoms at the interface between the top and bottom layers of bilayer $MoS_2$. From the compliance tensor analysis explained earlier, the strain along the zigzag direction ($\sigma_1 = \sigma$ and otherwise zero), for example, induces an interlayer shift along the armchair direction (Fig. 1d). Such a layer sliding along the armchair direction breaks the symmetry in the interface hexagonal lattice similar to the distorted intralayer hexagonal lattice under strain along the armchair direction (Fig. 1c). Thus, the effective bond distance between interface sulfur atoms parallel to the strain direction is shortened, while the perpendicular one is elongated as schematically illustrated in Fig. 1. This is indeed opposite to the

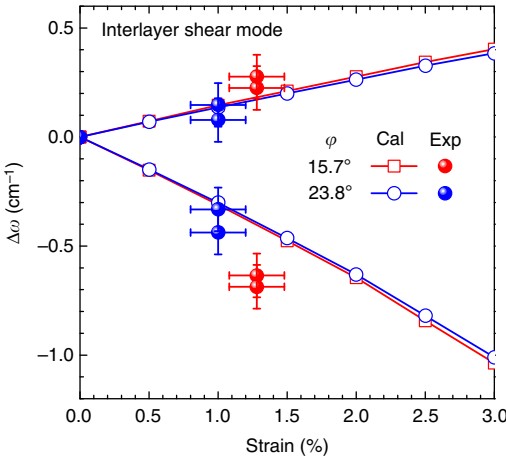

**Fig. 2** Splitting of interlayer shear mode under uniaxial strain. Shifts of split interlayer shear modes under uniaxial tensile strain for two different strain directions of $\varphi = 15.7°$ (red) and $23.8°$ (blue). Calculation (open circle and square) and experimental (filled symbols) results are compared. For the experimental data, values obtained from the Stokes and anti-Stokes spectra are plotted. Experimental error bars in strain come from the uncertainty in the estimate of the strain using the equation $\varepsilon = t_0\theta/L_0$ as explained in Methods. The error bars in $\Delta\omega$ are defined by the standard deviation of experimental values

in-plane bond elongation within a single layer under the same strain. Thus, one can expect that the polarization dependence of Raman scattering intensity of the hardened shear mode should be similar to that of softened intralayer optical phonon modes and vice versa. From our first-principles calculations, we have also checked that the frequency splitting for the case of uniaxial stretching without sliding. It has shown that the splitting becomes ten times smaller compared to the splitting when the sliding is properly considered (see Supplementary Fig. 2). We developed a detailed interface phonon model using effective interatomic interaction between the sulfur atoms at the interface and obtain a quantitative expression for the splitting of shear phonon modes under strain agreeing well with our first-principles calculation results (Supplementary Note 1 and Supplementary Fig. 3). From all theoretical considerations, we expect that the anomalous splitting of shear phonon modes under uniaxial strain should occur irrespective of direction of applied in-plane strain.

**Raman spectroscopic determination of shear mode splitting.** We have performed polarized Raman experiments to compare with our theoretical prediction. Bilayer $MoS_2$ samples are prepared directly on acrylic substrates by mechanical exfoliation from $MoS_2$ flakes (Supplementary Fig. 4). The number of layers is determined by combination of optical contrast, Raman and photoluminescence (PL) measurements. Experimental details of application of strain and polarized ultralow-frequency Raman measurements are presented in "Methods" section.

Figure 3a shows the Raman spectra of bilayer $MoS_2$ for different uniaxial strains up to 1.88%. The $A_{1g}$ and $E_g$ modes of bilayer $MoS_2$ correspond to intralayer vibrations, the interlayer in-plane mode (shear mode, $S$) corresponding to $E_g$ and the interlayer out-of-plane $A_{1g}$ mode (breathing mode, $B$) are also observed (see Supplementary Fig. 4a for schematics of vibration modes)[14,15]. First, we focus on the intralayer high-frequency modes under strain. The in-plane $E_g$ mode redshifts and splits into two peaks as the strain increases, whereas the $A_{1g}$ mode does not vary much[7–10]. Since the $A_{1g}$ mode is an out-of-plane mode, the effect of in-plane strain is minimal[7–10]. The $E_g$ mode splits

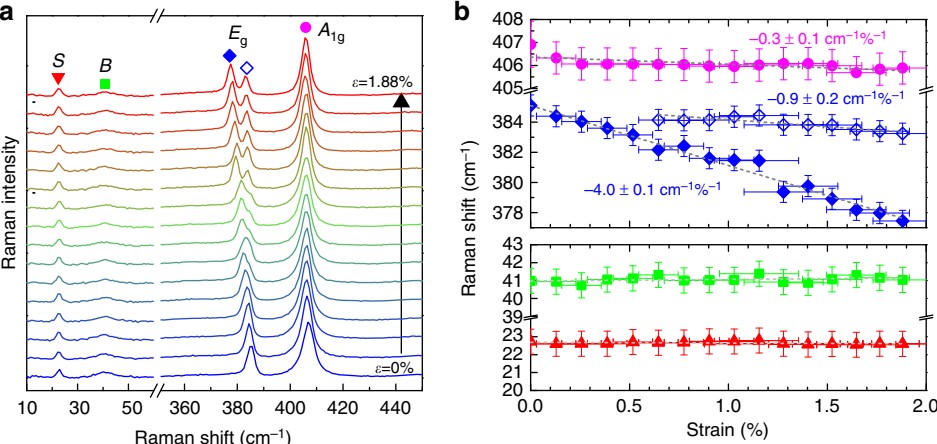

**Fig. 3** Raman spectra of bilayer MoS$_2$ under uniaxial strain. **a** Raman spectra of bilayer MoS$_2$ as a function of uniaxial strain. From left to right, the shear, breathing, split $E_g$, and $A_{1g}$ modes are denoted by the triangle, square, diamond, and circle symbols, respectively. **b** The peak positions for intralayer phonon modes (upper panel) and interlayer shear and breathing modes (lower panel) as a function of uniaxial tensile strain. Error bars in strain come from the uncertainty in the estimate of the strain using the equation $\varepsilon = t_0 \theta / L_0$. The error bars in Raman shift are defined by spectral resolution of the spectrometer

owing to the symmetry breaking of the hexagonal lattice[4,7,10,32]. We label the split mode with lower (higher) frequency as $E_g^-$ ($E_g^+$). We note that since the strain lowers symmetry of the system, the vibration modes of the strained MoS$_2$ are no longer $E_g$[8,28]. Nevertheless, we label the split modes as $E_g^-$ and $E_g^+$ in order to indicate their origin. The overall behavior of intralayer Raman modes of bilayer MoS$_2$ is similar to that of the single-layer case[7,10]. The interlayer shear ($S$) and breathing ($B$) modes show negligible dependence on strain. These results are summarized in Fig. 3b. Assuming linear dependence of the frequencies on the uniaxial strain, the shift rates of the $E_g^-$, $E_g^+$, and $A_{1g}$ modes for bilayer are obtained to be $-4.0 \pm 0.1$, $-0.9 \pm 0.2$, and $-0.3 \pm 0.1$ cm$^{-1}$%$^{-1}$, respectively, agreeing well with previous studies[7]. We compared the experimental data with calculations with different in-plane Poisson's ratio (see Supplementary Fig. 1). The experimental results seem to fit best with calculations with the Poisson's ratio of intrinsic bilayer MoS$_2$ ($\nu_i = 0.22$)[33], which we used for the following analysis. From the shift rates, the Grüneisen parameter ($\gamma$) and the shear deformation potential ($\beta$) for the $E_g$ modes are calculated as $\gamma = 1.0 \pm 0.2$ and $\beta = 0.8 \pm 0.2$, which are also similar to previous results[7] (Supplementary Note 3). The polarization dependences of the in-plane and out-of-plane intralayer modes with strain are shown in Fig. 4a and summarized in Fig. 4b, c, respectively. By inspecting the polarized Raman spectrum of in-plane intralayer modes, we can determine the direction of strain with respect to the crystal orientation following the well-established procedure[4] (Supplementary Note 4). Analyzing the data in Fig. 4b, we obtain the crystallographic orientation of the sample, $\varphi = 15.7 \pm 1.1°$. For another sample, we obtained $\varphi = 23.8 \pm 2.1°$ (Supplementary Fig. 5).

Now we turn our focus onto the interlayer shear modes. Figure 5a, b compare the polarization dependence of the low-frequency Raman modes without and with uniaxial strain, respectively. The intensity of the breathing mode ($B$) shows a dependence on the polarization similar to the case of the $A_{1g}$ mode irrespective of strain. Under strain, the shear mode peak intensity does not seem to depend on polarization, and no apparent splitting is observed at any given polarization. Upon close inspection, however, one can notice that the shear mode peak at ~22.8 cm$^{-1}$ seems to move periodically with polarization unlike the unstrained case as shown in Fig. 5a, b. Since the phonon mode frequency itself should not depend on polarization, the apparent polarization-dependent shift can only be interpreted

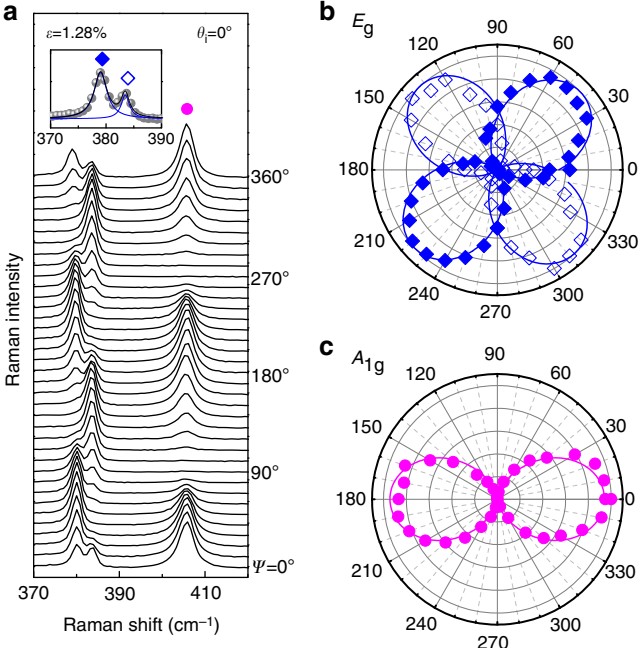

**Fig. 4** Polarization dependence of intralayer Raman modes under strain. **a** Polarized Raman spectra of strained bilayer MoS$_2$ ($\varepsilon = 1.28\%$) for main intralayer modes. The incident polarization ($\theta_i$) is fixed at 0° and the spectra are measured as a function of $\psi$ defined in Fig. 1a. Empty and filled diamond and circle symbols are defined in Fig. 3. Normalized polar plots of **b** $E_g$ ($E_g^-$ and $E_g^+$) and **c** $A_{1g}$ modes

as being due to polarization dependence of the relative intensities of closely spaced peaks due to strain-induced splitting. This kind of a small splitting can be distinguished with a help of polarization[34,35] because degeneracy-lifted phonon modes have different polarization dependences[4,6,10,32]. Since the peaks are not resolved well at any particular polarization angle, a simple double Lorentzian fitting of individual spectrum would not be able to determine the splitting reliably. Therefore, we measured a whole set of 37 spectra as a function of polarization at fixed strain of 1.28%. Since the frequency of the modes should not depend on the polarization, the whole set of 37 spectra were fitted as a whole by requiring that the peak positions are the same in all 37 spectra

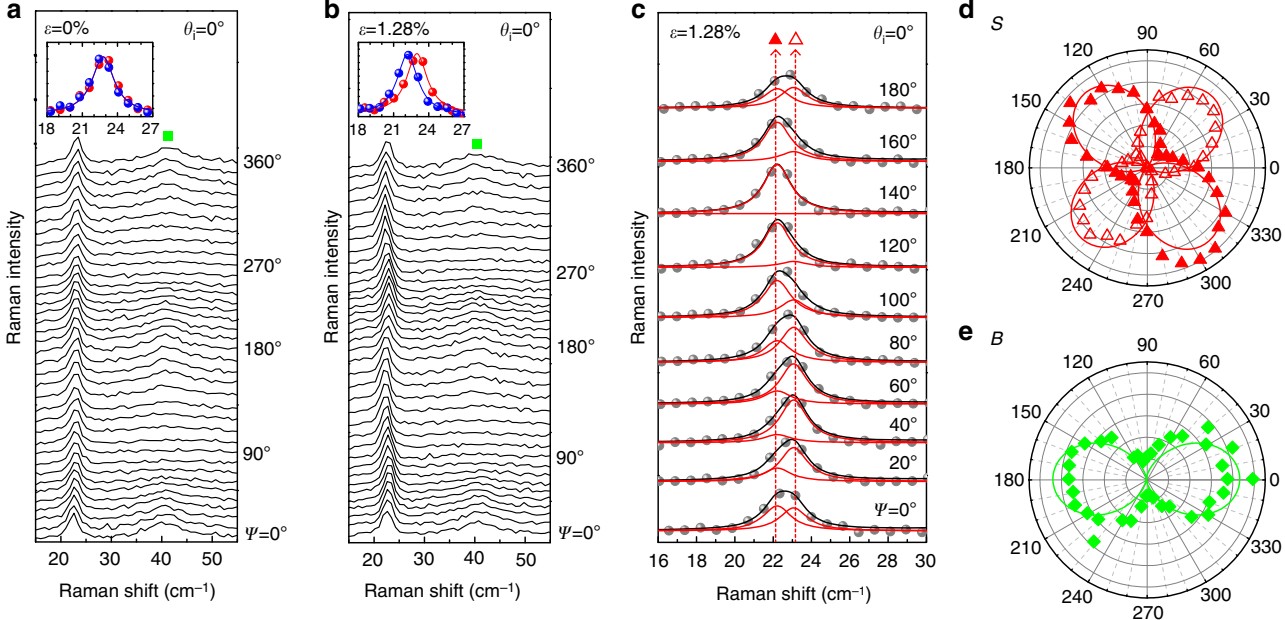

**Fig. 5** Polarization dependence of interlayer Raman modes. Polarized Raman spectra of bilayer MoS$_2$ **a** without and **b** with strain ($\varepsilon = 1.28\%$) in the low-frequency region. The incident polarization ($\theta_i$) is fixed at 0° and the spectra are measured as a function of $\psi$ defined in Fig. 1a. The insets compare spectra taken for $\psi = 40°$ (blue) and 130° (red) and show the splitting for the case with strain. **c** Representative fitting results for split shear modes. Normalized polar plots of **d** shear ($S$, $S^-$, and $S^+$) and **e** breathing ($B$) modes

**Table 1 Elastic constants of bilayer MoS$_2$**

| Elastic constants | Value | References |
|---|---|---|
| $s_{14}$ | $-1.46 \pm 0.34$ TPa$^{-1}$ (exp)/ $-0.84$ TPa$^{-1}$ (cal) | This work |
| $s_{44}$ | $111 \pm 8$ TPa$^{-1}$ (exp)[a]/ $154$ TPa$^{-1}$ (cal) | This work |
| $E_i$ | 0.33 TPa (exp) | Ref. [36] |
| $E_o$ | $0.058 \pm 0.002$ TPa (exp)[b] | This work |
| $\nu_i$ | 0.22 (cal) | Ref. [33] |
| $\nu_o$ | 0.18 (cal) | Ref. [33] |

Note: Summary of elastic constants estimated from experimental results
[a]Experimental value of $s_{44}$ was estimated from $c_{44}$, obtained from the position of the (unstrained) shear modes[14,15], assuming that off-diagonal elements are much smaller than diagonal elements ($s_{44} \approx 1/c_{44}$)
[b]Experimental value of $E_o$ was obtained from the position of the (unstrained) breathing mode[14,15]

and only the relative intensities vary (Fig. 5c). By doing so, one can greatly reduce the experimental uncertainty in the positions of the two peaks as shown in Fig. 5c and find that the peak splitting is ~$0.91 \pm 0.05$ cm$^{-1}$ under 1.28% uniaxial strain. The validity of this procedure is ascertained by Fig. 5d, in which the intensities of the split peaks obtained through the fitting procedure follow the dependences expected from strain-split peaks of $E_g$ phonon modes (Supplementary Note 4).

For the interlayer shear modes, we denote the low-frequency modes and high-frequency modes as $S^-$ and $S^+$, respectively, as mentioned earlier. The polarization dependence of the shear and breathing modes are shown in Fig. 5d, e. We also find almost similar shifts in other samples with different strain directions (Supplementary Fig. 5). The experimentally obtained peak positions under strain are plotted in Fig. 2, which show good agreement with our first-principles calculations. As expected from our theoretical analysis, the splitting behavior of the interlayer shear modes is indeed in sharp contrast to the intralayer case in that the split mode in the direction parallel to

the uniaxial strain hardens while the other mode in the direction perpendicular to it softens with strain. The polarization dependence of $S^+$ (Fig. 5d) is the same as that of $E_g^-$ (Fig. 4b), and vice versa, confirming our theoretical analysis. The corresponding Grüneisen parameter and shear deformation potentials for the interlayer shear phonon are obtained to be $0.9 \pm 0.3$ and $2.4 \pm 0.4$, respectively.

### Discussion

An off-diagonal elastic modulus $s_{14}$ can be obtained by comparing the experimental data with the model based on the interface hexagonal sulfur networks. With a simple spring model between the interface sulfur atoms, the amount of frequency splitting can be estimated to be $\Delta\omega = (\omega_0 d_{int} \gamma' |s_{14}| E_i / 2k)\varepsilon$, where $\omega_0$, $d_{int}$, $E_i (= 1/s_{11})$, and $\varepsilon$ are the phonon frequency without strain, the interlayer distance, the in-plane Young's modulus, and the strain along the applied uniaxial stress, respectively; $k$ is the lateral component of the effective spring constant between nearest-neighbor interface sulfur atoms from different layers; $\gamma'$ is the linear scaling factor of $k$ to the change of interatomic distance from the equilibrium; and $s_{14}$ is the element of the compliance tensor between $\varepsilon_1$ and $\sigma_4$ (see Supplementary Note 1 for derivation of $\Delta\omega$). On the other hand, the average shift $\overline{\omega}$ of the two split frequencies is given by $\Delta\overline{\omega} = \overline{\omega} - \omega_0 = -(1 - \nu_i)(\omega_0 \gamma' a_{SS}/4k)\varepsilon$ (see Supplementary Note 1 for derivation), where $a_{SS}$ is the interlayer sulfur-to-sulfur lateral distance, and $\nu_i$ is the in-plane Poisson's ratio. Combining the expressions for $\Delta\omega$ and $\Delta\overline{\omega}$, we obtain $|s_{14}| = -(a_{SS}/d_{int})((1 - \nu_i)/2E_i)\Delta\omega/\Delta\overline{\omega}$. With $a_{ss} = 0.18$ nm, $d_{int} = 0.62$ nm, $\nu_i = 0.22$, $E_i = 0.33$ TPa[33,36,37] and the experimental values of $\Delta\omega/\varepsilon = 0.60 \pm 0.13$ cm$^{-1}$%$^{-1}$, and $\Delta\overline{\omega}/\varepsilon = -0.15 \pm 0.04$ cm$^{-1}$%$^{-1}$, we find that $|s_{14}|$ of MoS$_2$ is $1.46 \pm 0.34$ TPa$^{-1}$. Most of the uncertainty comes from a small $\Delta\overline{\omega}$ value that goes into the denominator. Within this uncertainty, our experimentally obtained $|s_{14}|$ value is consistent with our first-principles calculation result of 0.84 TPa$^{-1}$, which is obtained following a procedure similar to that in ref. [33]. From these considerations, we tabulate all relevant elastic parameters in Table 1, and the complete matrix

elements of compliance tensor in Eq. (1) for bilayer $MoS_2$ can be written as:

$$\begin{pmatrix} \varepsilon_1 \\ \varepsilon_2 \\ \varepsilon_3 \\ \varepsilon_4 \\ \varepsilon_5 \\ \varepsilon_6 \end{pmatrix} = \begin{pmatrix} 3.0 & -0.67 & -0.55 & -1.46 & 0 & 0 \\ -0.67 & 3.0 & -0.55 & 1.46 & 0 & 0 \\ -0.55 & -0.55 & 17 & 0 & 0 & 0 \\ -1.46 & 1.46 & 0 & 111 & 0 & 0 \\ 0 & 0 & 0 & 0 & 111 & 2.92 \\ 0 & 0 & 0 & 0 & 2.92 & 7.4 \end{pmatrix} \begin{pmatrix} \sigma_1 \\ \sigma_2 \\ \sigma_3 \\ \sigma_4 \\ \sigma_5 \\ \sigma_6 \end{pmatrix},$$

(2)

where the matrix elements are in the unit of $TPa^{-1}$. Here, we used experimental values wherever available.

By analyzing the low-frequency Raman spectrum, we have shown that the hitherto unexplored off-diagonal elastic constant $s_{14}$ of $MoS_2$ can be estimated. Our study here can be easily extended to other multilayered 2D crystals and could open a new way to determine almost all elastic constants of layered materials.

## Methods

**Computational methods.** For the theoretical analysis, we have carried out first-principles calculations using Quantum ESPRESSO package[38] with plane wave basis and norm-conserving pseudopotentials[39]. To include the interlayer van der Waals interaction properly, we used the revised version[40] of the nonlocal correlation functional method developed by Vydrov and van Voorhis[41]. Phonon frequencies are calculated using density functional perturbation theory[38,42]. The energy cutoff for the basis set expansion is 110 Ry. Such a high-energy cutoff is adopted for resolving the low frequencies of interlayer shear modes which is order of a few meV. The $k$-point grid of $8 \times 8 \times 1$ is used. In order to obtain $s_{14}$, we have alternatively calculated $s_{56}$ ($= -2s_{14}$) by inverting the matrix form of the shear part of stiffness tensor $c_{ijkl}$, where i, j, k, l $= x, y, z$. The stiffness tensor is defined by $c_{ijkl} = \partial^2 F / \partial \varepsilon_{ij} \partial \varepsilon_{kl}$. Here, $F$ is the total free energy of the system from the first-principles calculations as a function of discrete values of strain $\varepsilon_{ij}$ up to 5%, and the differentiation is done numerically. The thickness of a bilayer $MoS_2$ is set to be twice the interlayer distance. For more details, refer to Supplementary Note 2.

**Experimental details of Raman measurements.** The bilayer samples of $MoS_2$ were prepared directly on acrylic substrates by mechanical exfoliation from $MoS_2$ flakes (SPI supplies). The number of layers was determined by the combination of optical contrast, Raman and PL measurements (Supplementary Fig. 4). The uniaxial strain is applied by a four-probe bending stage and calculated by $\varepsilon = t_0 \theta / L_0$, where $L_0$ and $t_0$ is unstrained length and thickness of acrylic substrate, $\theta$ is an angle in arc by bending substrate[6–8]. The laser beam was focused onto the sample by a 50× microscope objective lens (0.8 N.A.), and the scattered light was collected and collimated by the same objective. The scattered signal was dispersed with a Jobin-Yvon Triax 550 spectrometer (1800 grooves $mm^{-1}$) and detected with a liquid-nitrogen-cooled back-illuminated charge-coupled device detector. To access the low-frequency range below 100 $cm^{-1}$, reflective volume holographic filters (Ondax) were used to reject the Rayleigh-scattered light. The spectral resolution of our system is ~0.7 $cm^{-1}$. The laser power was kept below 0.2 mW in order to avoid heating.

**Data availability.** The data that support the findings of this study are available from the corresponding author upon request.

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

## Acknowledgements

This work was supported by the National Research Foundation (NRF) grant funded by the Korean government (MSIP) (NRF-2016R1A2B3008363 and No. 2017R1A5A1014862, SRC program: vdWMRC center) and by a grant (No. 2011-0031630) from the Center for Advanced Soft Electronics under the Global Frontier Research Program of MSIP. Computations were supported by the Center for Advanced Computation of Korea Institute for Advanced Study.

## Author contributions

H.C. and Y.-W.S. supervised the project. S.W. and H.C.P. performed first-principles calculations and model analysis. J.-U.L. and J.P. performed Raman spectroscopy measurements and data analysis. Y.-W.S. and H.C. co-wrote the paper and J.-U.L. and S.W. commented on the manuscript. All authors discussed the results and manuscript at all stages.

## Additional information

**Competing interests:** The authors declare no competing financial interests.

