## [Peer Review File · Nature Communications]

Reviewers' comments:

Reviewer #1 (Remarks to the Author):

The manuscript reports the use of Raman spectroscopy measurements to study the vibrational modes of bilayer MoS₂ when submitted to a uniaxial strain up to 1.88%. The authors claim that the strain induces the anomalous splitting of the degenerate interlayer shear mode, and that the mode along the strain direction is hardened, while the shear mode along the direction perpendicular to this strain is softened, contrary to the commonly found for the intralayer modes. They use theoretical analysis to understand the connection of the uniaxial strain with the shear in the system, which implies a lateral sliding of the upper layer with respect to the lower layer, finding a value for the off-diagonal elastic modulus S_{14} . It is important to notice that this is a new concept to be introduced in the "flatland". Density Functional Perturbation Theory was also used to estimate the values of the shift rates of the two S^{+} and S^{-} shear modes.

In page 4, line 10 (P4L10 from now on), the authors say: 'Our results could therefore provide a way to measure a whole set of elastic constants of layered systems that characterize their mechanical properties completely'. Please include in your results section a table with all the elastic constants you used in your modeling (yours and from literature). It will be highly clarifying to the readership and will give the broad view expected due to the previous assumption.

In P6L10 the authors say: 'The shifts do not depend sensitively on the strain direction'. Here we have only two experimental measurements, with data acquired at the experimental limit resolution as the authors say in the manuscript. It seems a strong affirmation. I suggest the authors to explain better or exclude this.

In Table 1 the first-principles calculations show the frequency shift of the interlayer shear modes of the system with respect to the uniaxial strain. Please explain why the two calculated strain angle ϕ are not the same as the ones used for the two experimental measurements. Is there any calculation difficulty?

In P6L14 the authors say: 'Our first-principles calculation also implies that the interlayer sliding induced by uniaxial strain is not negligible at all'. Please give the value and direction of this sliding.

In P7L9 the authors say: 'We also confirm from our first principles calculation that uniaxial stretching of both layers without sliding does not reproduce any splitting of the shear modes'. This assumption is counterintuitive because the lowering in symmetry naturally imposed by strain is expected to break the frequency degeneracy of these vibrational modes. Yet, strain applied to different directions is expected to leave a bilayer with different symmetries, with lower space groups in which these modes would not be degenerate. The exotic frequency value strain dependence could be accounted by the change in the S atoms distance, but the fundamental nature of this splitting would not depend exclusively from this effect. Please comment this issue in your manuscript and give the exact frequency value calculated from first principles for this mode. What was the strain angle in this case? How many atoms exist in the bilayer unit cell for this symmetry? Is there another reference with a similar result?

In the experimental details of Raman measurements, please include your experimental resolution.

By analyzing the novelty of this research, the quality of the manuscript, the scope and the standard expected to Nature Communications, the manuscript met the expected criteria. It is an interesting reference to a broad readership, especially those working aiming the architecture of new materials electrical and mechanical properties and development, characterization and device fabrication, in which electron-phonon interactions can play a role. The theoretical study of the off-diagonal elastic modulus can motivate others to study its effects in real applications and in other materials. Therefore, I feel that this manuscript is suitable for publication at this journal after the

correction of the suggested modifications.

Reviewer #2 (Remarks to the Author):

I have found this paper conceptually interesting but with the major limitation of the uncertainty of the final key result, i.e. the evaluation of the s_{14} elastic constant: $2.5 \pm 2.0 \text{ TPa}^{-1}$ (vs the first principle calculation of 0.8 TPa). This uncertainty should be reduced by one order of magnitude for "demonstrating that Raman spectroscopy can determine almost all mechanical – I would say elastic – constants of layered materials" considering also s_{44} ! (and strictly speaking also other 2D materials). Please provide details on the ab initio simulations and the numerical determination of the entire compliance tensor. I am not an expert in Raman spectroscopy and thus I must assume these measurements well done. The evaluation of s_{14} requires a model that could perhaps be improved, e.g. I am not convinced about the validity of all the related assumptions, such as the linear dependence of the stiffness, the still quadratic formal dependence of the potential energy, the self-consistency of the substitution of the stiffness with their nonlinear counterparts (line 70; where the Poisson ratio value of 0.4 of the "substrate" (?) is used?), etc. Thus I suggest the authors to try to reduce the uncertainty of s_{14} and evaluate also s_{44} and validate the entire compliance tensor with ab initio simulations.

Reviewer #3 (Remarks to the Author):

The authors report a comprehensive study on the effect of uniaxial strain on the low frequency Raman modes in bilayer MoS_2 . An interface hexagonal model is put forward to explain the shear mode splitting under strain. Comparing the strain induced splitting of the high frequency E_{2g}^{+1} mode by polarized Raman scattering, the authors justify their model by the same method of measuring the two sub-peaks of shear mode (S^{+} & S^{-}).

The authors do not seem to aware of the two very relevant studies by Li et al. (Journal of Solid State Science and Technology 5 Q3033 (2016)) and Huang et al. (Nano Lett. 16 1435 (2016)). These two studies have addressed the low frequency modes in MoS_2 .

Besides, there are some critical problems in the manuscript.

(1) The authors explained that the splitting of shear modes would solely depend on the way of breaking the hexagonal symmetry. It means that the symmetry between the adjacent interface sulfur atoms play an important role in the calculation. The authors only consider bilayer MoS_2 . As the stacking order has great influence on the symmetry, can the authors' model extend to multilayer MoS_2 ?

(2) The authors' interface model considers the bond length only. However, a realistic model should also take the changes in both bond length and bond angles into consideration. (Computational Materials Science 44 1142 (2009) & The Journal of Physical Chemistry C 121 5366 (2017)). Apparently, the changes of bond angles are observed in Figure 1.

(3) The authors chose Poisson's ratio as 0.4 for the acrylic substrate to replace the Poisson's ratio of MoS_2 . Other groups used 0.33 (Nano Lett. 13 3626 (2013), PRB 79 205433 (2009)), 0.35 (PRB 87 081307(R)) and 0.16 (PNAS 106 7304) for different 2D materials. What is the reason of selecting the Poisson's ratio of 0.4 for their system? Typically a polymer interlayer is coated on the substrate to ensure the good adhesion between the sample and the substrate so that no slippage will occur under strain. How could the authors ensure no slippage occurred in their sample under a strain as large as 1.8%?

(4) The data for the E_{g+} mode under uniaxial strain from 0 % to 0.5 % seems missing in Figure 2(b). Please check it.

(5) Horiba Triax 550 with 1800 g/mm was used to acquire the Raman signals. Based on my understanding, the spectrum resolution is about 0.6 cm^{-1} in this condition. The authors claimed that the largest peak splitting is $\sim 0.64 \pm 0.1 \text{ cm}^{-1}$. The value is very close to the resolution of system. Moreover, the wavenumber interval of the adjacent point as shown in Figure 4 (c) is less than 1 cm^{-1} . Therefore, the observed peak splitting may not be reliable and it could be due to the system error.

(6) The authors mainly discussed the shear mode of the sample. Why does the breathing mode show two-fold symmetry like the A_{1g} mode in the polarized Raman spectrum? The underlying basic physics is not described.

The manuscript is not warrant to publish in Nature Communications in the present form.

Reviewer #4 (Remarks to the Author):

The authors investigate the elastic constants of MoS₂ through Raman measurements including shear mode, breathing mode, E_g mode and A_{1g} mode of bilayer MoS₂ under strain together with simulated results of shear mode. The results are meaningful to the research community of mechanical properties of layered materials. However, the novelty of this work may not meet the requirements of the journal of Nature communications. Moreover, there are still some concerns which need to be further clarified. Therefore, I recommend that a major revision is needed before any further consideration.

My comments and concerns are as follows.

(1) As said on page 9 that "...the maximum of amount of peak splitting ($\sim 0.64 \pm 0.1 \text{ cm}^{-1}$) is quite close to our resolution limit.". What is the spectral resolution of the Raman system? Please give a detailed explanation of the spectral resolution by measuring atomic emission line of Mercury.

(2) For Figure 2 and Figure 4(b), it is hard to tell at which strain the shear mode starts to split. Could you show the width of the shear mode by fitting by one Lorentz peak as a function of strain? If the shear mode starts to split, there should be a broadening of the shear mode. It is farfetched to say the shear mode splits by the intensities of the fitted peaks follow the dependences expected from E_g modes. Please reconsider this.

(3) In Figure S3(b), the range of X-axis should be $370 - 390 \text{ cm}^{-1}$.

(4) Could you discuss the importance and significance of off-diagonal elastic constant s_{14} ?

Authors' Response to the Reviewers' Remarks:

We thank the referees for careful reading of our manuscript and helpful suggestions. In order to address their comments, we have repeated the calculations with higher accuracy and analyzed the experimental data with more rigor. As a result, the theoretical prediction and the experimental data agree better and the conclusions are on much firmer grounds. In the revised version, we added several figures and references to support our conclusions. We explain the revisions in the following point-by-point answers to the reviewers' comments.

Reviewer #1

Remarks to the authors

The manuscript reports the use of Raman spectroscopy measurements to study the vibrational modes of bilayer MoS₂ when submitted to a uniaxial strain up to 1.88%. The authors claim that the strain induces the anomalous splitting of the degenerate interlayer shear mode, and that the mode along the strain direction is hardened, while the shear mode along the direction perpendicular to this strain is softened, contrary to the commonly found for the intralayer modes. They use theoretical analysis to understand the connection of the uniaxial strain with the shear in the system, which implies a lateral sliding of the upper layer with respect to the lower layer, finding a value for the off-diagonal elastic modulus S_{14} . It is important to notice that this is a new concept to be introduced in the "flatland". Density Functional Perturbation Theory was also used to estimate the values of the shift rates of the two S^{+} and S^{-} shear modes.

Authors' Response

We thank the referee for careful reading of our manuscript and encouraging comments.

Remarks to the authors

In page 4, line 10 (P4L10 from now on), the authors say: ‘Our results could therefore provide a way to measure a whole set of elastic constants of layered systems that characterize their mechanical properties completely’. Please include in your results section a table with all the elastic constants you used in your modeling (yours and from literature). It will be highly clarifying to the readership and will give the broad view expected due to the previous assumption.

Authors’ Response

We thank the referee for the suggestion. We have tabulated the elastic constants in Table 1 and the numerical values of the compliance tensor components are given in Eq. (2).

Remarks to the authors

In P6L10 the authors say: ‘The shifts do not depend sensitively on the strain direction’. Here we have only two experimental measurements, with data acquired at the experimental limit resolution as the authors say in the manuscript. It seems a strong affirmation. I suggest the authors to explain better or exclude this.

Authors’ Response

We thank the referee for the good comment. The statement is based mainly on the first-principles calculations as well as analytic expressions given in Supplementary Information. The experimental results also support the statement. In our revised version, we have included the shift of phonon frequencies from first-principles calculations corresponding to the experimental directions of strain, 23.8° and 15.7° . They are shown in Fig. 2 as a function of strain. It shows a negligible difference in the frequency shifts between the two strain directions (and other cases with two angles reported in our previous manuscript also show the same shift).

TEXT CHANGES

(Page 6, Line 10) “The shifts do not depend sensitively on the strain direction.”

→ (Page 6, Line 16) “Our first-principles calculations show that the shifts barely depend on the strain direction (See Fig. 2 and Supplementary Fig. 2).”

Remarks to the authors

In Table 1 the first-principles calculations show the frequency shift of the interlayer shear modes of the system with respect to the uniaxial strain. Please explain why the two calculated strain angle ϕ are not the same as the ones used for the two experimental measurements. Is there any calculation difficulty?

In P6L14 the authors say: 'Our first-principles calculation also implies that the interlayer sliding induced by uniaxial strain is not negligible at all'. Please give the value and direction of this sliding.

Authors' Response

We thank the referee for comments. We have recalculated the phonon frequencies for the strain angles corresponding to the experimental ones and substituted Table 1 with Fig. 2. The sliding amplitude is almost independent of the direction of the applied strain. It is about 1.5 % of the lattice constant of the primitive unit cell when 3 % of uniaxial strain is applied. The directions of sliding as a function of strain direction from the first-principles calculations match our model prediction (solid line) very well as shown in the figure below. The x and y axes denote the strain direction and the sliding direction, respectively. The solid squares are the data points from our previous version and the hollow ones are the data points from our current version that correspond to the experimental strain directions.

TEXT CHANGES

(Page 6, Line 14) “Our first-principles calculation also implies that the interlayer sliding induced by uniaxial strain is not negligible at all.”

→ (Page 6, Line 20) “The amount of strain-induced interlayer sliding based on the first-principles calculation is not negligible at all and is about a half percent of the lattice constant for the applied strain of one percent. It is also confirmed that the direction of sliding matches well with our model prediction.”

Remarks to the authors

In P7L9 the authors say: ‘We also confirm from our first principles calculation that uniaxial stretching of both layers without sliding does not reproduce any splitting of the shear modes’. This assumption is counterintuitive because the lowering in symmetry naturally imposed by strain is expected to break the frequency degeneracy of these vibrational modes. Yet, strain applied to different directions is expected to leave a bilayer with different symmetries, with lower space groups in which these modes would not be degenerate. The exotic frequency value strain dependence could be accounted by the change in the S atoms distance, but the fundamental nature of this splitting would not depend exclusively from this effect. Please comment this issue in your manuscript and give the exact frequency value calculated from first principles for this mode. What was the strain angle in this case? How many atoms exist in the bilayer unit cell for this symmetry? Is there another reference with a similar result?

Authors' Response

We thank the referee for insightful comments. We have rechecked our calculations and have found that the splitting does exist even without sliding as the referee pointed out. The splitting without sliding, however, is very small compared to one with sliding. It is only about 10 % of the total splitting with induced interlayer sliding (See Supplementary Figure 2). It means that only 10 % of splitting is due to the deformation of the hexagonal symmetry and the major part of the splitting is due to sliding. Therefore, it is reasonable to take the sliding as the leading-order contribution for the observed splitting. In Supplementary Fig. 2 of the revised

manuscript, we have included the first-principles data of interlayer shear mode frequencies of strained MoS₂ without sliding for proper comparison.

In our previous version we chose 0°, 5°, 14°, 30° for the strain angles for first-principles calculations. In the revised version we have used the experimental parameters instead, 15.7° and 23.8°. We have used a primitive unit cell so that there are six atoms in a bilayer unit cell. The strain is simulated by modifying the unit vectors. As far as we know, there has been no previous research for the strain dependence of the interlayer shear phonon frequencies for bilayer MoS₂. The phonon frequencies of single-layer and bulk MoS₂ were compared in PRL 84, 155413 (2011).

TEXT CHANGES

(Page 7, Line 9) “We also confirm from our first principles calculation that uniaxial stretching of both layers without sliding does not reproduce any splitting of the shear modes, thereby justifying our interface model.”

→ (Page 7, Line 17) “From our first-principles calculations, we have also checked that the frequency splitting for the case of uniaxial stretching without sliding. It has shown that the splitting becomes ten times smaller compared to the splitting when the sliding is properly considered (See Supplementary Fig. 2).”

Remarks to the authors

In the experimental details of Raman measurements, please include your experimental resolution.

Authors' Response

We thank the referee for pointing out the omission. We have added a sentence in the section of Experimental details of Raman measurements.

TEXT ADDITION

(Page 14, Line 2) “The spectral resolution of our system is $\sim 0.7 \text{ cm}^{-1}$.”

Remarks to the authors

By analyzing the novelty of this research, the quality of the manuscript, the scope and the standard expected to Nature Communications, the manuscript met the expected criteria. It is an interesting reference to a broad readership, especially those working aiming the architecture of new materials electrical and mechanical properties and development, characterization and device fabrication, in which electron-phonon interactions can play a role. The theoretical study of the off-diagonal elastic modulus can motivate others to study its effects in real applications and in other materials. Therefore, I feel that this manuscript is suitable for publication at this journal after the correction of the suggested modifications.

Authors' Response

We thank the referee for the support.

Reviewer #2

Remarks to the authors

I have found this paper conceptually interesting but with the major limitation of the uncertainty of the final key result, i.e. the evaluation of the s_{14} elastic constant: $2.5\text{-}+2.0\text{ TPa}^{-1}$ (vs the first principle calculation of 0.8TPa). This uncertainty should be reduced by one order of magnitude for “demonstrating that Raman spectroscopy can determine almost all mechanical – I would say elastic – constants of layered materials” considering also s_{44} ! (and strictly speaking also other 2D materials). Please provide details on the ab initio simulations and the numerical determination of the entire compliance tensor. I am not an expert in Raman spectroscopy and thus I must assume these measurements well done. The evaluation of s_{14} requires a model that could perhaps be improved, e.g. I am not convinced about the validity of all the related assumptions, such as the linear dependence of the stiffness, the still quadratic formal dependence of the potential energy, the self-consistency of the substitution of the stiffness with their nonlinear counterparts (line 70; where the Poisson ratio value of 0.4 of the “substrate” (?) is used?), etc. Thus I suggest the

authors to try to reduce the uncertainty of s14 and evaluate also s44 and validate the entire compliance tensor with ab initio simulations.

Authors' Response

We thank the referee for careful reading of our manuscript and encouraging comments. The computational methods for the first principles calculations are described in the Methods section of the main text and the details of the ab initio calculations for the compliance tensor is now added in Supplementary Information as Supplementary Note 2. In the revised version, we have added Fig. 2 that shows the linear dependence of frequency splitting on the applied strain. It shows that the range of strain used for our theory and experiment is well within the linear regime.

Regarding the use of Poisson's ratio of the substrate, we had assumed that the MoS₂ sample on the substrate does not experience any slippage. However, for more direct evidence, we compared the calculated shifts of the strain-split high-frequency Eg modes using several values of Poisson's ratio: 0, 0.22 (intrinsic Poisson's ratio of MoS₂), and 0.4 (Poisson's ratio of the substrate). It turned out that 0.22 fits best with the experimental data (See Supplementary Figure 1). Therefore, we have revised all the analysis with the Poisson's ratio value of 0.22.

TEXT CHANGES

(Page 11, Line 3) “On the other hand, the shift of the average $\bar{\omega}$ of the two split frequencies is given by $\Delta\bar{\omega} = \bar{\omega} - \omega_0 = -(1 - \nu_i)(\omega_0 \gamma' a_{ss} / 4k)\epsilon$ (see Supplementary Note 1 for derivation), where a_{ss} is the interlayer sulfur-to-sulfur lateral distance, and ν_i is the actual in-plane Poisson's ratio applied on MoS₂ which is that of the substrate in the experiment.”

→ (Page 9, Line 4) “We compared the experimental data with calculations with different in-plane Poisson's ratio (See Supplementary Fig. 1). The experimental results seem to fit best with calculations with the Poisson's ratio of intrinsic bilayer MoS₂ ($\nu_i = 0.22$)³³, which we used for the following analysis.”

(Page 7, Line 17) “On the other hand, the average shift $\bar{\omega}$ of the two split frequencies is given by $\Delta\bar{\omega} = \bar{\omega} - \omega_0 = -(1 - \nu_i)(\omega_0 \gamma' a_{ss} / 4k)\epsilon$ (see Supplementary Note 1 for

derivation), where a_{ss} is the interlayer sulfur-to-sulfur lateral distance, and ν_i is the in-plane Poisson's ratio.”

As for the error bars in the estimated value of s_{14} , we corrected an error in data analysis and also improved the statistics of the data by including the data from both Stokes and anti-Stokes spectra. We found that we made an error by using the lattice constant instead of the sulfur-to-sulfur distance for a_{ss} in the previous version. Using the correct a_{ss} reduced the s_{14} value. Also, by including both Stokes and anti-Stokes data, we have 4 sets of experimental data instead of 2, which reduced the error bars significantly. The new estimate of s_{14} is $-1.46 \pm 0.34 \text{ TPa}^{-1}$, which is much improved in terms of uncertainty and the agreement with theoretical calculations.

Whereas we could obtain the compliance tensor elements directly from first-principles calculations, one needs to invert the stiffness tensor c_{ij} to estimate it experimentally. Since we do not know all the values of c_{ij} , it is not possible to obtain s_{44} rigorously. On the other hand, if we assume that the unknown elements are 0 (which is supported by first-principles calculations), we obtain $s_{44} = 111 \pm 8 \text{ TPa}^{-1}$, which compares with the calculated value of 154 TPa^{-1} . We now include this value in Table 1 with proper caveat. The complete compliance tensor is now given as Eq. (2).

Reviewer #3

Remarks to the authors

The authors report a comprehensive study on the effect of uniaxial strain on the low frequency Raman modes in bilayer MoS₂. An interface hexagonal model is put forward to explain the shear mode splitting under strain. Comparing the strain induced splitting of the high frequency E_{2g1} mode by polarized Raman scattering, the authors justify their model by the same method of measuring the two sub-peaks of shear mode (S⁺ & S⁻).

The authors do not seem to aware of the two very relevant studies by Li et al. (Journal of Solid State Science and Technology 5 Q3033 (2016)) and Huang et al.

(Nano Lett. 16 1435 (2016)). These two studies have addressed the low frequency modes in MoS₂.

Authors' Response

We thank the referee for careful reading of our manuscript, and suggestions of relevant references. We have added the references in our revised version as Refs. 16 and 17.

Remarks to the authors

Besides, there are some critical problems in the manuscript.

(1) The authors explained that the splitting of shear modes would solely depend on the way of breaking the hexagonal symmetry. It means that the symmetry between the adjacent interface sulfur atoms play an important role in the calculation. The authors only consider bilayer MoS₂. As the stacking order has great influence on the symmetry, can the authors' model extend to multilayer MoS₂?

Authors' Response

We thank the referee for the insightful comments. For MoS₂ with more than 2 layers, the stacking is such that the interlayer shear effect alternates the sign across the layers. As a result, the interlayer sliding effect cancels out as the number of layer increases. In other words, one can say that the extreme 2D materials such as a bilayer MoS₂ is a system that maximizes the shear mode splitting.

Remarks to the authors

(2) The authors' interface model considers the bond length only. However, a realistic model should also take the changes in both bond length and bond angles into consideration. (Computational Materials Science 44 1142 (2009) & The Journal of Physical Chemistry C 121 5366 (2017)). Apparently, the changes of bond angles are observed in Figure 1.

Authors' Response

Figure 1 is a schematic that exaggerates the deformation, and the actual angle distortion is very small. Furthermore, in a deformed hexagonal lattice made of

covalent bonds, the energy cost due to the bond length change is expected to be dominant over that due to the bond angle distortion. For the case of carbon network, for example, in PRB 61, 10651 (L. M. Woods and G. D. Mahan, 'Electron-phonon effects in graphene and armchair (10,10) single-wall carbon nanotubes'), the bond angle potential is almost 10 times smaller than the bond length potential. We believe that it is fair to explain our experiment using the dominant factor first.

Remarks to the authors

(3) The authors chose Poisson's ratio as 0.4 for the acrylic substrate to replace the Poisson's ratio of MoS₂. Other groups used 0.33 (Nano Lett. 13 3626 (2013), PRB 79 205433 (2009)), 0.35 (PRB 87 081307(R)) and 0.16 (PNAS 106 7304) for different 2D materials. What is the reason of selecting the Poisson's ratio of 0.4 for their system? Typically a polymer interlayer is coated on the substrate to ensure the good adhesion between the sample and the substrate so that no slippage will occur under strain. How could the authors ensure no slippage occurred in their sample under a strain as large as 1.8%?

Authors' Response

This Remark is essentially identical to the one by Reviewer #2. We repeat our response below.

Regarding the use of Poisson's ratio of the substrate, we had assumed that the MoS₂ sample on the substrate does not experience any slippage. However, for more direct evidence, we compared the calculated shifts of the strain-split high-frequency E_g modes using several values of Poisson's ratio: 0, 0.22 (intrinsic Poisson's ratio of MoS₂), and 0.4 (Poisson's ratio of the substrate). It turned out that 0.22 fits best with the experimental data (See Supplementary Figure 1). Therefore, we have revised all the analysis with the Poisson's ratio value of 0.22.

TEXT CHANGES

(Page 11, Line 3) "On the other hand, the shift of the average $\bar{\omega}$ of the two split frequencies is given by $\Delta\bar{\omega} = \bar{\omega} - \omega_0 = -(1 - \nu_t)(\omega_0 \gamma' a_{ss} / 4k)\epsilon$ (see Supplementary Note 1 for derivation), where a_{ss} is the interlayer sulfur-to-sulfur lateral distance, and

ν_i is the actual in-plane Poisson's ratio applied on MoS₂ which is that of the substrate in the experiment.”

→ (Page 9, Line 4) “We compared the experimental data with calculations with different in-plane Poisson's ratio (See Supplementary Fig. 1). The experimental results seem to fit best with calculations with the Poisson's ratio of intrinsic bilayer MoS₂ ($\nu_i = 0.22$)³³, which we used for the following analysis.”

(Page 7, Line 17) “On the other hand, the average shift $\bar{\omega}$ of the two split frequencies is given by $\Delta\bar{\omega} = \bar{\omega} - \omega_0 = -(1 - \nu_i)(\omega_0 \gamma' a_{ss} / 4k)\epsilon$ (see Supplementary Note 1 for derivation), where a_{ss} is the interlayer sulfur-to-sulfur lateral distance, and ν_i is the in-plane Poisson's ratio.”

Remarks to the authors

(4) The data for the Eg+ mode under uniaxial strain from 0 % to 0.5 % seems missing in Figure 2(b). Please check it.

Authors' Response

As one can see from Figure 2(a), the splitting of the Eg mode is very small for small strain between 0 and 0.5%. The data points (filled diamonds) for this range should be considered the convoluted average peak position of the two split modes. Although such a small splitting could have been estimated by performing polarized Raman measurements as we did for the low-frequency shear mode, we did not do it because the splitting of the Eg mode is not the focus of this work.

Remarks to the authors

(5) Horiba Triax 550 with 1800 g/mm was used to acquire the Raman signals. Based on my understanding, the spectrum resolution is about 0.6 cm⁻¹ in this condition. The authors claimed that the largest peak splitting is $\sim 0.64 \pm 0.1$ cm⁻¹. The value is very close to the resolution of system. Moreover, the wavenumber interval of the adjacent point as shown in Figure 4 (c) is less than 1 cm⁻¹. Therefore, the observed peak splitting may not be reliable and it could be due to the system error.

Authors' Response

There is an error in the previous version: $0.64 \pm 0.1 \text{ cm}^{-1}$ is the splitting per 1 percent of strain and the maximum splitting measured is 0.91 cm^{-1} for 1.28% strain. Nevertheless, the referee is correct that the system resolution ($\sim 0.7 \text{ cm}^{-1}$) and the splitting (0.91 cm^{-1} for 1.28% strain) that we are trying to measure are of similar order. However, with ingenious statistical techniques one can overcome the limit is the system resolution by employing enough statistics to give much better confidence in the peak position with an uncertainty much smaller than the system resolution. This is commonly done in single-molecule imaging and super-resolution microscopy, which won the Nobel Prize in chemistry in 2014.

In our case, instead of trying to estimate the peak positions from individual spectrum, we treated the whole set of 37 spectra as one data set. Since the spectrometer does not move between measurements, any systematic shift or drift can be excluded. Since the phonon mode frequency should not depend on the polarization, the simultaneous fitting of the 37 spectra was carried out by requiring that the positions of the two peaks do not change depending on the polarization and only the relative intensity changes. This procedure greatly improves the statistical confidence on the estimate of the two peak positions. The insets in Figures 5(a) and (b) of the revised manuscript clearly show that the splitting exists only for the strained case.

We realize that the description in the previous version was not very clear. We have modified the text in the revised version to explain the procedure more clearly.

TEXT CHANGES

(Page 9, Line 16) “Therefore, we measured a whole set of 37 spectra as a function of polarization at fixed strain of 1.28% and globally fitted the entire set of spectra with the same parameter except for the intensity (Fig. 4c). By doing so, we can clearly identify two peaks as shown in Fig. 4c and find that the maximum of amount of peak splitting ($\sim 0.64 \pm 0.1 \text{ cm}^{-1}$) is quite close to our resolution limit.”

→ (Page 10, Line 7) “Therefore, we measured a whole set of 37 spectra as a function of polarization at fixed strain of 1.28%. Since the frequency of the modes should not depend on the polarization, the whole set of 37 spectra were fitted as a whole by requiring that the peak positions are the same in all 37 spectra and only the relative

intensities vary (Fig. 5c). By doing so, one can greatly reduce the experimental uncertainty in the positions of the two peaks as shown in Fig. 5c and find that the peak splitting is $\sim 0.91 \pm 0.05 \text{ cm}^{-1}$ under 1.28% uniaxial strain.”

Remarks to the authors

(6) The authors mainly discussed the shear mode of the sample. Why does the breathing mode show two-fold symmetry like the A_{1g} mode in the polarized Raman spectrum? The underlying basic physics is not described.

The manuscript is not warrant to publish in Nature Communications in the present form.

Authors' Response

Because the vibration of the breathing mode is in the direction normal to the strain direction, it is not affected much by the strain. This is the same with the A_{1g} mode. In fact, since the breathing mode and the A_{1g} mode belong to the same irreducible representation, the polarization behavior should be the same as shown in Fig. 4(c) and Fig. 5(e).

Reviewer #4

Remarks to the authors

The authors investigate the elastic constants of MoS₂ through Raman measurements including shear mode, breathing mode, E_g mode and A_{1g} mode of bilayer MoS₂ under strain together with simulated results of shear mode. The results are meaningful to the research community of mechanical properties of layered materials. However, the novelty of this work may not meet the requirements of the journal of Nature communications. Moreover, there are still some concerns which need to be further clarified. Therefore, I recommend that a major revision is needed before any further consideration.

Authors' Response

We thank the referee for careful reading of our manuscript and encouraging comments.

Remarks to the authors

My comments and concerns are as follows.

(1) As said on page 9 that "...the maximum of amount of peak splitting ($\sim 0.64 \pm 0.1$ cm⁻¹) is quite close to our resolution limit.". What is the spectral resolution of the Raman system? Please give a detailed explanation of the spectral resolution by measuring atomic emission line of Mercury.

Authors' Response

We thank the Reviewer for the good suggestion. The system resolution is 0.7 cm⁻¹, which is now added to the main text.

TEXT ADDITION

(Page 14, Line 2) "The spectral resolution of our system is ~ 0.7 cm⁻¹."

On the other hand, the uncertainty in the determination of the splitting is much better than the system resolution because of ample statistics. This point is explained in response to Reviewer #3's comment (5). We copy our response below.

There is an error in the previous version: 0.64 ± 0.1 cm⁻¹ is the splitting per 1 percent of strain and the maximum splitting measured is 0.91 cm⁻¹ for 1.28% strain. Nevertheless, the referee is correct that the system resolution (~ 0.7 cm⁻¹) and the splitting (0.91 cm⁻¹ for 1.28% strain) that we are trying to measure are of similar order. However, with ingenious statistical techniques one can overcome the limit is the system resolution by employing enough statistics to give much better confidence in the peak position with an uncertainty much smaller than the system resolution. This is commonly done in single-molecule imaging and super-resolution microscopy, which won the Nobel Prize in chemistry in 2014.

In our case, instead of trying to estimate the peak positions from individual spectrum, we treated the whole set of 37 spectra as one data set. Since the spectrometer does

not move between measurements, any systematic shift or drift can be excluded. Since the phonon mode frequency should not depend on the polarization, the simultaneous fitting of the 37 spectra was carried out by requiring that the positions of the two peaks do not change depending on the polarization and only the relative intensity changes. This procedure greatly improves the statistical confidence on the estimate of the two peak positions. The insets in Figures 5(a) and (b) of the revised manuscript clearly show that the splitting exists only for the strained case.

We realize that the description in the previous version was not very clear. We have modified the text in the revised version to explain the procedure more clearly.

TEXT CHANGES

(Page 9, Line 16) “Therefore, we measured a whole set of 37 spectra as a function of polarization at fixed strain of 1.28% and globally fitted the entire set of spectra with the same parameter except for the intensity (Fig. 4c). By doing so, we can clearly identify two peaks as shown in Fig. 4c and find that the maximum of amount of peak splitting ($\sim 0.64 \pm 0.1 \text{ cm}^{-1}$) is quite close to our resolution limit.”

→ (Page 10, Line 7) “Therefore, we measured a whole set of 37 spectra as a function of polarization at fixed strain of 1.28%. Since the frequency of the modes should not depend on the polarization, the whole set of 37 spectra were fitted as a whole by requiring that the peak positions are the same in all 37 spectra and only the relative intensities vary (Fig. 5c). By doing so, one can greatly reduce the experimental uncertainty in the positions of the two peaks as shown in Fig. 5c and find that the peak splitting is $\sim 0.91 \pm 0.05 \text{ cm}^{-1}$ under 1.28% uniaxial strain.”

Remarks to the authors

(2) For Figure 2 and Figure 4(b), it is hard to tell at which strain the shear mode starts to split. Could you show the width of the shear mode by fitting by one Lorentz peak as a function of strain? If the shear mode starts to split, there should be a broadening of the shear mode. It is farfetched to say the shear mode splits by the intensities of the fitted peaks follow the dependences expected from Eg modes. Please reconsider this.

Authors' Response

As the first-principles calculations clearly show in Figure 2 of the revised manuscript, the splitting increases almost linearly to the applied strain. Therefore, the peaks should start to split as soon as strain is applied, although the splitting cannot be resolved experimentally.

At the suggestion of the referee, we fitted the shear mode with a single Lorentzian function to see if the width shows broadening. As one can see below, the width indeed broadens gradually although the overall increase is small since the splitting itself is not very large.

Remarks to the authors

(3) In Figure S3(b), the range of X-axis should be 370 – 390 cm⁻¹.

Authors' Response

Thank you for pointing out our error. It is now fixed.

Remarks to the authors

(4) Could you discuss the importance and significance of off-diagonal elastic constant s_{14} ?

Authors' Response

The electronic properties such as band structures or band topology will be affected critically by the interlayer shear sliding when mechanical deformation is applied. This is especially so for low dimensional materials. So if one wants to develop devices based on especially 2D materials, it is important to consider such shear sliding. Since s_{14} is the most critical elastic constant linked to the relative sliding of the layers, it would play an important role when one considers electronic properties under strain such as in a flexible or stretchable device made of this material.

REVIEWERS' COMMENTS:

Reviewer #1 (Remarks to the Author):

The authors answered reasonably all the requests and that the manuscript is in good shape to be published.

Reviewer #2 (Remarks to the Author):

The authors have revised the manuscript, correcting the calculation and thus improving the num vs theo-exp comparison and providing an estimation of the entire compliance tensor. Now I suggest to accept this strengthened version of their paper.

Reviewer #3 (Remarks to the Author):

The authors have addressed some of the concerns raised by the referees. As part of the results presented in the current manuscript have already been studied by two separated groups (Li et al., Journal of Solid State Science and Technology 5 Q3033 2016 and Huang et al., Nano Lett. 16 1435 2016), the manuscript does not seem to provide new insight to the topic.

Reviewer #4 (Remarks to the Author):

The authors try to answer all the comments and questions raised by the reviewers in the response letter and the revised manuscript. It can be seen that the major comments are about the uncertainty and reliability of the calculations and experimental data analysis. Moreover, there are many mistakes the authors have realized regarding both the theoretical calculations and experimental data analysis in the original submitted version after questioned by the reviewers. Thus, I doubt more the validity and scientific rigour of present work. That is my opinion about this manuscript. I believe more serious considerations and revisions must be very helpful to further improve the work.

Authors' Response to the Reviewers' Remarks:

We thank the referees for another round of careful review of our manuscript. We have addressed the comment of Reviewer #3 by adding a phrase in our final revision.

Reviewer #1

Remarks to the authors

The authors answered reasonably all the requests and that the manuscript is in good shape to be published.

Authors' Response

We thank the referee for the kind comments.

Reviewer #2

Remarks to the authors

The authors have revised the manuscript, correcting the calculation and thus improving the num vs theo-exp comparison and providing an estimation of the entire compliance tensor. Now I suggest to accept this strengthened version of their paper.

Authors' Response

We thank the referee for the kind comments.

Reviewer #3

Remarks to the authors

The authors have addressed some of the concerns raised by the referees. As part of the results presented in the current manuscript have already been studied by two separated groups (Li et al., Journal of Solid State Science and Technology 5 Q3033

2016 and Huang et al., Nano Lett. 16 1435 2016), the manuscript does not seem to provide new insight to the topic.

Authors' Response

At the suggestion of this referee, We added the two papers as references in the previous version. The first one [Ref. 16] deals with low-frequency Raman modes of 3 to 13 layer MoS₂. The strain effects were measured on a sample with a thickness of 8.71 nm. Furthermore, there was no theoretical analysis of the strain effect in this paper. On the other hand, our measurements and theoretical analysis are specifically for 2-layer MoS₂ where the coupling between in-plane uniaxial strain and interlayer shear are most prominent. The other paper [Ref. 17] deals with twisted bilayer MoS₂ and has nothing to do with the strain effect. For clarification, we added a sentence in the introduction section.

TEXT CHANGES

(Page 3, Line 11) “Similarly, the effects of interlayer interactions on mechanical properties can be analyzed by probing the low-frequency interlayer shear and breathing modes¹³⁻¹⁷.”

→ (Page 3, Line 11) “Similarly, the effects of interlayer interactions on mechanical properties can be analyzed by probing the low-frequency interlayer shear and breathing modes¹³⁻¹⁷. Li *et al.* also reported the effect of strain on low-frequency interlayer modes in relatively thick (8.71 nm) MoS₂.¹⁶”

Reviewer #4

Remarks to the authors

The authors try to answer all the comments and questions raised by the reviewers in the response letter and the revised manuscript. It can be seen that the major comments are about the uncertainty and reliability of the calculations and experimental data analysis. Moreover, there are many mistakes the authors have realized regarding both the theoretical calculations and experimental data analysis in the original submitted version after questioned by the reviewers. Thus, I doubt more the validity and scientific rigour of present work. That is my opinion about this

manuscript. I believe more serious considerations and revisions must be very helpful to further improve the work.

Authors' Response

We take the criticism of the referee seriously. We are grateful that the errors were found and fixed through the rigorous review process.